# A kinase-independent function of AKT promotes cancer cell survival

Igor Vivanco[1]*[†‡], Zhi C Chen[1†], Barbara Tanos[2], Barbara Oldrini[3], Wan-Ying Hsieh[4], Nicolas Yannuzzi[1], Carl Campos[1], Ingo K Mellinghoff[1,4,5]*

[1]Human Oncology and Pathogenesis Program, Memorial Sloan-Kettering Cancer Center, New York, United States; [2]Cell Biology Program, Memorial Sloan-Kettering Cancer Center, New York, United States; [3]Seve Ballesteros Foundation Brain Tumor Group, Spanish National Cancer Research Centre, Madrid, Spain; [4]Department of Pharmacology, Weill-Cornell Graduate School of Biomedical Sciences, New York, United States; [5]Department of Neurology, Memorial Sloan-Kettering Cancer, New York, United States

**Abstract** The serine–threonine kinase AKT regulates proliferation and survival by phosphorylating a network of protein substrates. In this study, we describe a kinase-independent function of AKT. In cancer cells harboring gain-of-function alterations in MET, HER2, or Phosphatidyl-Inositol-3-Kinase (PI3K), catalytically inactive AKT (K179M) protected from drug induced cell death in a PH-domain dependent manner. An AKT kinase domain mutant found in human melanoma (G161V) lacked enzymatic activity in vitro and in AKT1/AKT2 double knockout cells, but promoted growth factor independent survival of primary human melanocytes. ATP-competitive AKT inhibitors failed to block the kinase-independent function of AKT, a liability that limits their effectiveness compared to allosteric AKT inhibitors. Our results broaden the current view of AKT function and have important implications for the development of AKT inhibitors for cancer.

*For correspondence: igor. vivanco@icr.ac.uk (IV); mellingi@ mskcc.org (IKM)

†These authors contributed equally to this work

Present address: ‡Male Urological Cancer Research Centre, Division of Cancer Therapeutics, Institute of Cancer Research, Sutton, United Kingdom

Competing interests: The authors declare that no competing interests exist.

## Introduction

The serine–threonine kinase AKT is a non-receptor kinase that plays critical roles in growth and metabolism (*Pearce et al., 2010*). AKT is aberrantly activated in many human cancers, usually due to mutations in upstream components of the Phosphatidyl-Inositol-3-Kinase (PI3K) pathway. AKT signaling is required for tumor maintenance in certain genetic contexts, as has been shown for some cancer cell lines harboring PI3K mutations or amplification of the HER2 receptor tyrosine kinase (*She et al., 2008*). Encouraged by these results, several ATP-competitive and allosteric AKT inhibitors have been identified (*Engelman, 2009*), but optimal strategies for the clinical development of these agents are currently unknown.

## Results

To address this question, we compared the activity of the allosteric AKT inhibitor MK2206 (*Hirai et al., 2010*) with the ATP-competitive AKT inhibitors GSK690693 (*Heerding et al., 2008*; *Rhodes et al., 2008*) and GDC0068 (*Lin et al., 2013*). Each of these AKT inhibitors dose dependently inhibited AKT kinase activity in the non-small-cell lung carcinoma (NSCLC) cell line EBC1 as determined by phosphorylation of three AKT substrates (PRAS40, GSK3β, and BAD) in whole-cell lysates (*Figure 1A* and *Figure 1—figure supplement 1*). Consistent with previous reports (*Okuzumi et al., 2009*), the ATP-competitive AKT inhibitors induced hyper-phosphorylation of AKT at the two regulatory phosphorylation sites threonine 308 and serine 473. When we compared the ability of the AKT inhibitors to induce cell death using a trypan blue exclusion assay, we observed significantly greater cell death induction

**eLife digest** To maintain a healthy body, the ability of our cells to survive and divide is normally strictly controlled. If any cells manage to escape these restrictions, they may rapidly divide and form tumors, which can lead to cancer.

A protein called AKT can encourage cells to survive and divide, and in healthy cells it is only allowed to be active at specific times. However, in many cancer cells, the genes that make and control AKT activity can be altered by mutations, which can result in AKT being active at the wrong times.

Part of the AKT protein acts as an enzyme called a kinase and adds chemical groups called phosphates to other proteins. The phosphate groups can activate or deactivate these proteins to control cell survival and cell division. However, there are other sections to the AKT protein and it is not clear how they are involved in this protein's activity.

In this study, Vivanco et al. show that AKT has another role in cell survival that does not depend on its kinase. The experiments show that even when the kinase part of the protein is missing, AKT can help cancer cells to survive drug treatments and external conditions that would normally kill them. This role requires another section of the protein called the PH-domain.

There are several chemicals—called inhibitors—that can stop AKT from working properly, and they have the potential to be used to treat some types of cancer. These inhibitors work in different ways: some were able to block the activity of the kinase, but others inhibited AKT by binding to other parts of the protein. Therefore, to develop AKT inhibitors into effective drugs, it will be important to know precisely what role the protein plays in different types of cancers.

by the allosteric inhibitor MK2206 compared to the ATP-competitive AKT inhibitors GSK690693 (*Figure 1B—source data 1*) and GDC0068 (*Figure 1C—source data 1*) despite less complete inhibition of AKT substrate phosphorylation at these drug concentrations (*Figure 1A*).

We had selected EBC1 lung cancer cells for our initial studies because they harbor amplification of *MET*, a receptor tyrosine kinase that mediates critical oncogenic signals through the PI3K/AKT axis (*Gherardi et al., 2012*). To test whether other *MET*-amplified human cancer cells show a similar drug response pattern, we extended our experiments to *MET*-amplified H1993 and H1648 NSCLC cell lines and GTL-16 gastric cancer cells. MK2206 again induced significantly more cell death than GSK690693 (*Figure 1—figure supplement 2*, *Figure 1—source data 1*). In cell lines with other PI3K pathway alterations previously associated with 'AKT-dependence' (*She et al., 2008*), we also found that MK2006 induced significantly more cell death than ATP-competitive AKT inhibitors. These cell lines included breast cancer cells with *HER2* amplification and/or *PIK3CA*-mutation (*Figure 1D* and *Figure 1—figure supplement 3*, *Figure 1—source data 1*). In contrast, MK2206 did not induce cell death in non-transformed bronchial epithelial cells *(data not shown)*. MK2206 also did not induce cell death in lung cancer cell lines with activating EGFR mutations (*Figure 1—figure supplement 4*, *Figure 1—source data 1*) which have been linked with aberrant PI3K pathway activation (*Engelman et al., 2005*), suggesting that the mode of PI3K activation is critical in establishing AKT addiction in cancer.

To exclude the possibility that the greater cell death induction by the allosteric AKT inhibitor was due to additional effects on proteins other than AKT, we stably expressed a synthetic AKT mutant in which a residue critical for allosteric inhibitor binding (tryptophan 80) (*Green et al., 2008*; *Calleja et al., 2009*; *Wu et al., 2010*) had been replaced by an alanine. Expression of AKT1 W80A or AKT2 W80A but not wild-type AKT1 or wild-type AKT2 completely rescued MK2206-induced cell death (*Figure 1E* and *Figure 1—figure supplement 5*, *Figure 1—source data 1*). Western blot analysis confirmed that these AKT alleles are biochemically insensitive to MK2206 and prevented the induction of PARP cleavage, a marker of apoptosis (*Figure 1—figure supplement 6*, *Figure 1—source data 1*).

Of note, all three inhibitors were able to induce near-complete proliferation arrest (*Figure 1—figure supplement 7*, *Figure 1—source data 1*). These results demonstrate that allosteric and ATP-competitive AKT inhibitors differ in their ability to block AKT-mediated survival, and that this survival signal can be mediated by AKT1 or AKT2.

AKT regulates survival through phosphorylation of several protein substrates. We were therefore surprised that ATP-competitive inhibitors induced less cell death than MK2206 despite greater or equal inhibition of AKT substrate phosphorylation (*Figure 1A* and *Figure 1—figure supplement 8*).

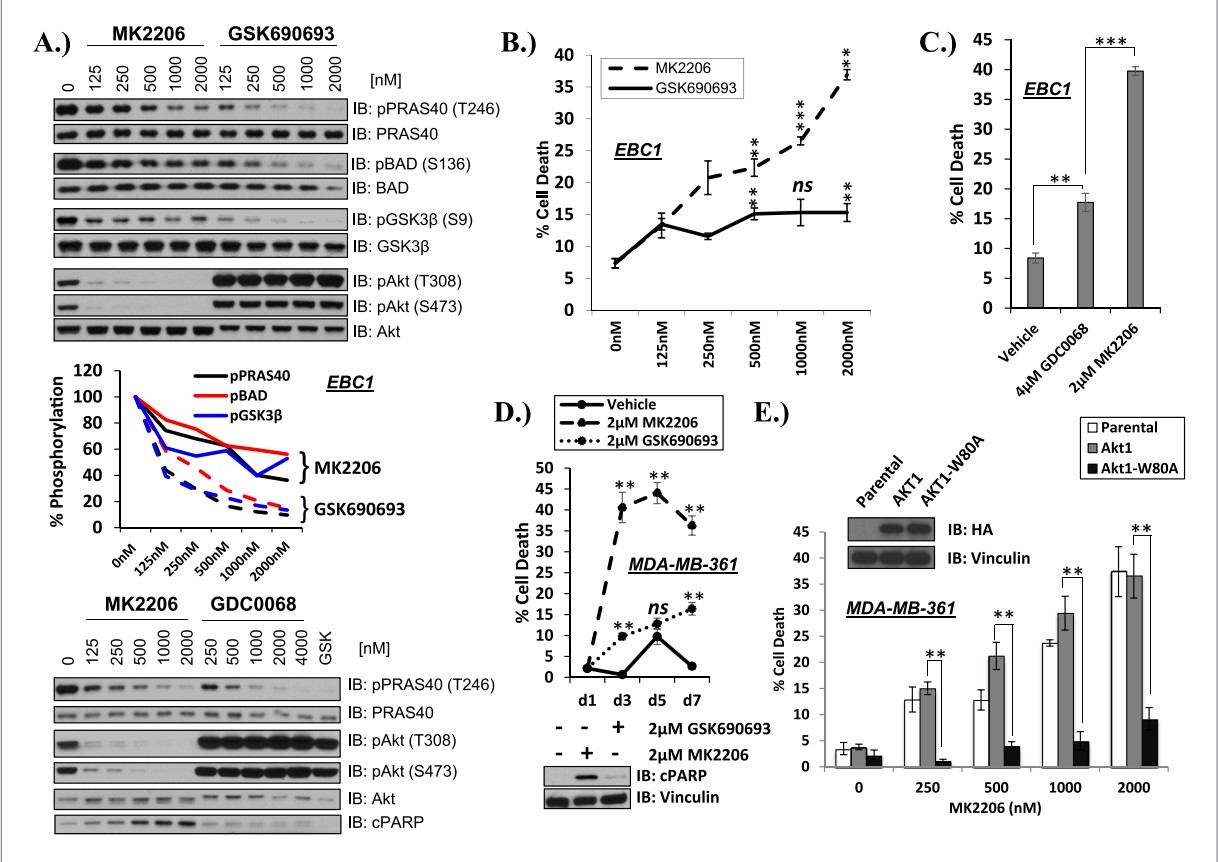

**Figure 1**. Differential sensitivity of human cancer cells to different classes of AKT inhibitors. (**A**) Effect of AKT inhibitors on phosphorylation of AKT and AKT protein substrates. EBC1 cells were treated with the indicated doses of MK2206 and GSK690693 (top), or GDC0068 (bottom) for 24 hr. Treated cells were lysed and analyzed by immunoblot with the indicated antibodies. Phosphorylation of PRAS40, BAD, and GSK3β was quantified by image densitometry (middle). (**B**) Allosteric AKT inhibitor MK2206 induces more cell death than ATP-competitive inhibitor GSK690693 in EBC1 human lung cancer cells. Cells were treated with drug or vehicle for 96 hr. Cell death was determined by the trypan blue method. Error bars denote standard error of the mean (t-test *p ≤ 0.05, **p ≤ 0.01, treatment vs vehicle). (**C**) Allosteric AKT inhibitor MK2206 induces more cell death than ATP-competitive inhibitor GDC0068. Experimental conditions were as in *Figure 1B*. (**D**) Allosteric AKT inhibitor MK2206 induces more cell death than the ATP-competitive inhibitor GSK690693 in MDA-MB-361 human breast cancer cells. Cell death was assessed on day 1, day 3, day 5, and day 7 following treatment with vehicle or 2 μM of MK2206 or 2 μM of GSK690693(top) (t-test *p ≤ 0.05,**p ≤ 0.01, treatment vs vehicle). An additional plate from each treatment group was lysed 24 hr after drug treatment and analyzed by Western blot with the indicated antibodies (bottom). (**E**) Suppression of MK2206-induced cell death by a drug-resistant allele of AKT1. MDA-MB-361 cells were stably transduced with HA-tagged wild-type AKT1 or AKT1-W80A and treated with 2 μM MK2206 for 96 hr. Cell death was assessed as above. Expression of the transgenes was confirmed by immunoblot using an HA antibody, and loading was controlled with a vinculin antibody (inset).

The following source data and figure supplements are available for figure 1:

**Source data 1**. Contains source data for *Figure 1* and all accompanying Figure 1—figure supplements.

**Figure supplement 1**. MK2206 and GSK690693 cause sustained suppression of AKT-dependent BAD phosphorylation.

**Figure supplement 2**. MET-amplified cancer cell lines die in response to an allosteric but not an ATP-competitive AKT inhibitor.

**Figure supplement 3**. Breast cancer cell lines with HER2 amplification and/or activating PIK3CA-mutations die in response to an allosteric but not an ATP competitive AKT inhibitor.

**Figure supplement 4**. Non-small-cell lung cancer (NSCLC) cell lines with EGFR gene amplification and/or activating EGFR mutations are resistant to cell death induction by AKT inhibitors.

*Figure 1. Continued on next page*

*Figure 1. Continued*

**Figure supplement 5**. A drug-resistant allele of AKT2 suppresses MK2206-induced cell death.

**Figure supplement 6**. W80A mutation renders AKT1 and AKT2 resistant to MK2206.

**Figure supplement 7**. Near-complete inhibition of tumor cell proliferation by both allosteric and ATP-competitive inhibitors of AKT.

**Figure supplement 8**. Comparison of AKT kinase inhibition by GSK690693 and MK2206.

**Figure supplement 9**. De-inhibition of ErbB3 and p-IGF-IRβ by GSK690693 and MK2206.

We could not explain these results by differential effects on induction of ErbB3 expression or IGF-IRβ phosphorylation (*Figure 1—figure supplement 9*) which can accompany AKT inhibition and mediate resistance to AKT inhibitors (*Chandarlapaty et al., 2011*).

One potential explanation for our observations is that ATP-competitive AKT inhibitors might induce an AKT conformation which can serve as a platform for a kinase-independent survival signal (*Figure 2A*). If true, ATP-competitive AKT inhibitors might antagonize cell death induction by a variety of cell death stimuli, for example, inhibition of the MET kinase in *MET*-amplified EBC1 cells. We found this to be the case. Despite inducing a modest amount of cell death as single agent, GSK690693 antagonized the ability of crizotinib to kill EBC1 cells. The allosteric AKT inhibitor MK2206, in contrast, did not compromise the ability of crizotinib to induce cell death (*Figure 2B*, *Figure 2—source data 1*).

To provide genetic evidence for the kinase-independent AKT function, we took advantage of a synthetic AKT mutant which is catalytically inactive due to substitution of lysine by methionine at position 179 at the ATP-binding site (*AKT K179M*) (*Franke et al., 1995*). When we expressed this mutant in MCF7 breast cancer cells, MDA-MB-361 breast cancer cells and EBC1 lung cancer cells, catalytically inactive AKT (*Figure 2C/D/E*, *Figure 2—source data 1*) but not wild-type AKT (*Figure 2—figure supplement 1*, *Figure 2—source data 1*), protected from MK2206-induced cell death. Compared to the complete cell death protection provided by the drug-resistant W80A-AKT allele (*Figure 1E*, *Figure 1—source data 1*), the protection by the K179M-AKT allele was only partial, consistent with the coexistence of kinase-independent and kinase-dependent AKT signals.

We also used an AKT knockdown strategy to minimize the effect of AKT overexpression on the stoichiometry of the AKT signaling complex. For this purpose, we first generated EBC1 cells with doxycycline-inducible human-specific shRNAs for both AKT 1 and AKT 2 and then expressed a kinase-deficient (AKT1-K179M) AKT1 cDNA of murine origin that is partially resistant to knockdown due to mismatches in the mouse sequence. Protein levels of the hairpin-resistant AKT allele after doxycycline induction were similar to the levels of endogenous AKT in parental EBC1 cells (*Figure 2F*, top, lanes 6 and 8 vs lanes 1 and 3). Similar to our results with pharmacological AKT blockade, catalytically inactive AKT protected from cell death following doxycycline treatment and AKT depletion of these cells (*Figure 2F*, bottom, *Figure 2—source data 1*).

We next examined the effects of AKT inhibitors on phosphoinositide (PI) binding. In the absence of growth factors, AKT is inactive because intermolecular interactions between the pleckstrin homology (PH) and kinase domain (KD) keep the kinase in a closed conformation (*Calleja et al., 2009*). Following growth factor stimulation, AKT shifts to an open conformation and becomes accessible to phosphorylation on threonine 308 and serine 473, a process that requires localization of AKT to the plasma membrane through the interaction with the phospholipid products of PI3K, phosphatidyl-inositol-3,4,5-trisphosphate [PtdIns $(3,4,5)P_3$], and PtdIns$(3,4)P_2$. Using a PI pull-down assay, we observed that MK2206 markedly impaired phosphoinositide binding (*Figure 3A*). This result is consistent with findings from structural studies that allosteric AKT inhibitors lock AKT in a closed conformation with its phospholipid binding site blocked by the kinase domain (*Wu et al., 2010*). In contrast to our findings with MK2206, incubation with the ATP-competitive inhibitor GSK690693 increased binding of AKT to PtdIns$(3,4)P_2$ and PtdIns $(3,4,5)P_3$ and this was associated with increased localization of AKT to the plasma membrane (*Figure 3B*).

We next explored the structural requirements for the pro-survival signal of catalytically inactive AKT. Similar to GSK690693-bound AKT, the AKT1-K179M (or AKT2-K181M) mutant was hyper-phosphorylated

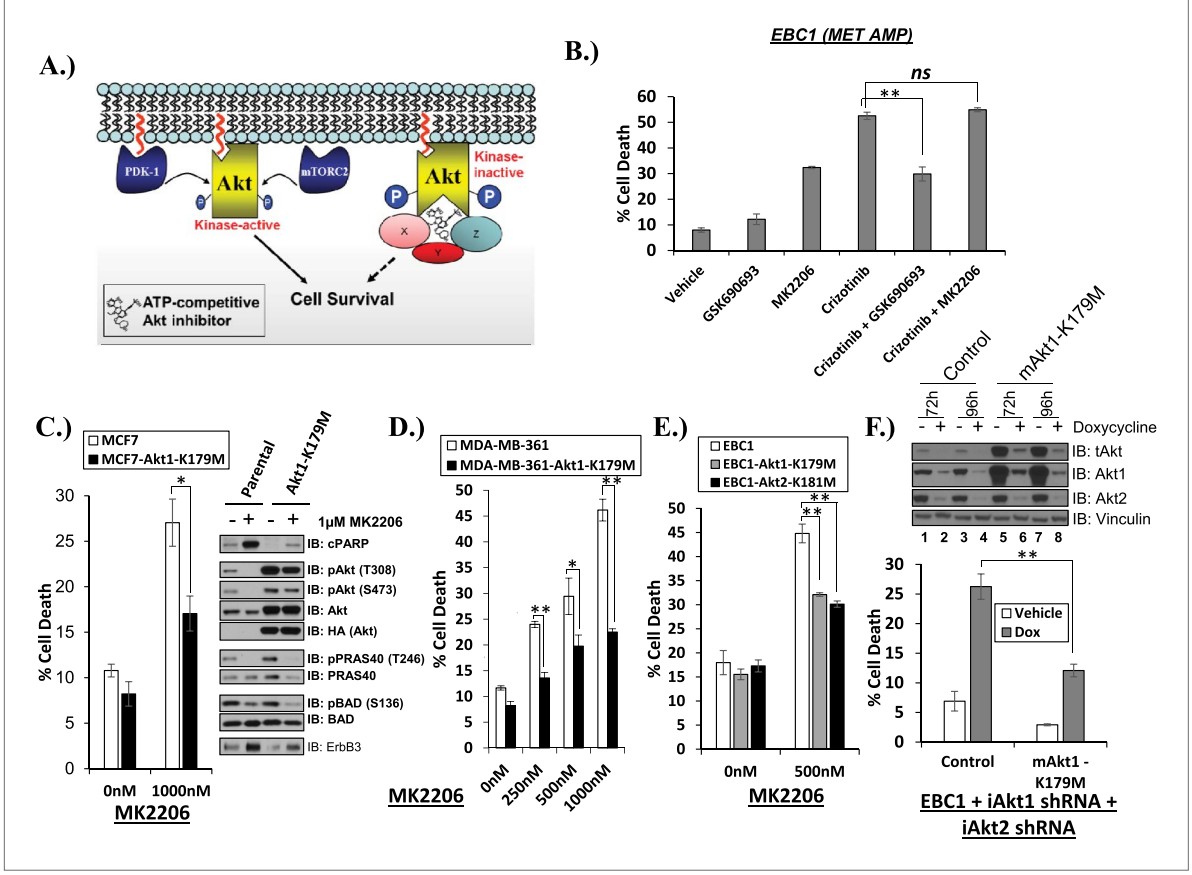

**Figure 2**. Catalytically inactive AKT supports the survival of AKT-dependent human cancer cells. (**A**) Cartoon illustrating how ATP-competitive AKT inhibitors may promote a kinase-independent pro-survival function of AKT. In this model, ATP-competitive inhibitors of AKT block kinase activity, induce AKT hyper-phosphorylation, and induce the formation of a pro-survival complex. (**B**) EBC1 cells were treated with either 25 nM crizotinib, 1 μM GSK690693, 1 μM MK2206, or the indicated combinations for 96 hr. Cell death was assessed following treatment by the trypan blue assay. (**C**) MCF7 cells or a sub-line generated by stable transduction with kinase-deficient AKT1-K179M were treated with either vehicle or 1 μM MK2206 for 96 hr. Cell death was assessed by the trypan blue method (left). A duplicate set of plates was lysed and analyzed by immunoblot with the indicated antibodies (right) to confirm kinase inhibition and transgene expression. (**D**) MDA-MB-361 cells or a sub-line stably expressing AKT1-K179M were treated with the indicated doses of MK2206 for 96 hr. Cell death was assessed by the trypan blue assay. (**E**) Parental EBC1 cells of EBC1 cells stably expressing AKT1-K179M or AKT2-K181M were treated with MK2206 as indicated. After 96 hr, cell death was assessed as elsewhere. (**F**) EBC1 cells expressing doxycycline-inducible hairpins targeting both hAKT1 and hAKT2 and a sub-line expressing a hairpin-resistant kinase-deficient mAKT1-K179M allele were grown in the presence or absence of 2.5 μg/ml doxycycline for 7 days. Cell death was assessed on day 7 by the trypan blue assay (bottom). Replicate plates were cultured with or without doxycycline for the indicated times and lysed. Lysates were analyzed by immunoblot (top) using the indicated antibodies.

The following source data and figure supplement are available for figure 2:

**Source data 1**. Contains source data for **Figure 2** and **Figure 2—figure supplement 1**.

**Figure supplement 1**. Kinase-dead AKT2-K181M but not wild-type AKT2 protects from MK2206-induced cell death.

at T308 (**Figure 3C** and **Figure 3—figure supplement 1**). This phosphorylation event occurs at the plasma membrane (**Stephens et al., 1998**) and immunofluorescence in MCF10A and EBC1 cells confirmed that kinase-dead AKT was constitutively localized to the plasma membrane (**Figure 3D** and **Figure 3—figure supplement 2**).

To examine whether membrane localization was sufficient for the pro-survival function of AKT1-K179M, we replaced the PH-domain of AKT1-K179M by a membrane-targeting myristoylation signal and expressed this mutant in EBC1 cells. This mutant was unable to inhibit MK2206-induced cell death (**Figure 3E**, **Figure 3—source data 1**) despite being constitutively phosphorylated in the presence of MK2206 (**Figure 3—figure supplement 3**). Lastly, we abolished the ability of the AKT1-K179M mutant

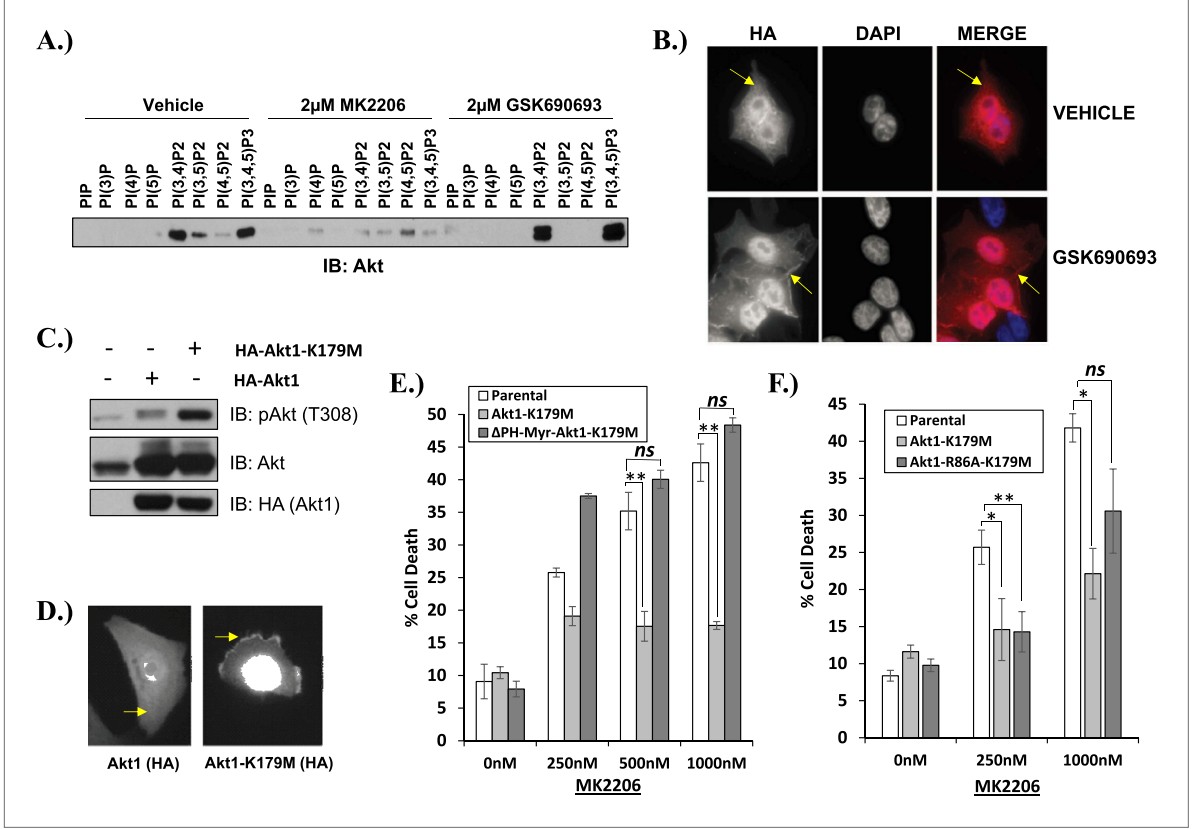

**Figure 3**. Kinase-dead AKT protects from drug-induced cell death in a PH-domain-dependent manner. (**A**) HCT116 AKT1−/− AKT2 −/− cells stably transduced with human influenza hemagglutinin (HA) epitope-tagged AKT2 were treated with vehicle, MK2206, or GSK690693 for 4 hr and lysed. Lysates were subjected to a PIP-binding assay to assess phosphoinositide binding preference. AKT binding was assessed by immunoblot using an AKT-specific antibody. (**B**) The localization of AKT2 was assessed in the same cells as in **A** by HA immunofluorescence following 24 hr treatment with the indicated drugs. Arrows indicate HA staining. (**C**) Parental MCF10A cells and MCF10A cells stably transduced with HA-tagged wild-type (WT) AKT1 (lane 2) or AKT1-K179M (lane 3) were analyzed by Western blot with the indicated antibodies. (**D**) The localization of WT and K179M-AKT1 in MCF10A cells was assessed by HA immunostaining. Shown are representative immunofluorescence images. Arrows indicate HA staining. (**E**) Parental MDA-MB-361 cells or sub-lines stably expressing AKT1-K179M or ΔPH-Myr-AKT1-K179M were treated with the indicated doses of MK2206 for 96 hr. Cell death was assessed by the trypan blue assay. (**F**) Parental MDA-MB-361 cells, or sub-lines stably expressing kinase-dead AKT1-K179M, or the PIP-binding-deficient variant AKT1-R86A-K179M were treated with the indicated doses of MK2206 for 96 hr. Cell death was measured by the trypan blue assay.

The following source data and figure supplements are available for figure 3:

**Source data 1**. Contains source data for **Figure 3** and **Figure 3—figure supplement 2**.

**Figure supplement 1**. Kinase-deficient AKT mutants display enhanced activation loop phosphorylation at threonine 308.

**Figure supplement 2**. Kinase-deficient mutant AKT2 exhibits enhanced membrane localization.

**Figure supplement 3**. Biochemical effects of PH-domain removal in kinase-dead AKT.

**Figure supplement 4**. Phosphoinositide binding deficient kinase-dead AKT can protect from drug-induced cell death.

to bind phospholipids by introducing a second substitution at an amino acid residue (R86) critical for binding of AKT to PtdIns (3,4,5)P$_3$ and PtdIns(3,4)P$_2$ (**Thomas et al., 2002**). Consistent with a loss of membrane localization, the R86A/K179M double mutant showed loss of AKT phosphorylation at threonine 308 when expressed in HCT116 cells that express no endogenous AKT (**Ericson et al., 2010**) (**Figure 3—figure supplement 4A**) and it did not protect from MK2206-induced dephosphorylation in cells which do express endogenous AKT (**Figure 3—figure supplement 4B**). Abrogation of

phosphoinositide binding by the R86A substitution did not impair the pro-survival function of kinase-defective AKT (*Figure 3F*, *Figure 3—source data 1*). We observed a similar result with a second mutation (R25C) known to abolish phosphoinositide binding (*Thomas et al., 2002*) (*Figure 3—figure supplement 4C*, *Figure 3—source data 1*).

Mutations in the coding sequence of AKT are rare in human cancers but provide important insights into AKT regulation (*Parikh et al., 2012*). We wondered whether there was precedence in human cancer for kinase-dead AKT mutations. Our survey of published cancer sequencing data identified five AKT2 kinase domain mutations that occurred in at least two independent samples (G161V/E, R170W, I289M, H355Y, R368C/H) (*Figure 4A*). We engineered each of these mutations into an AKT2 expression

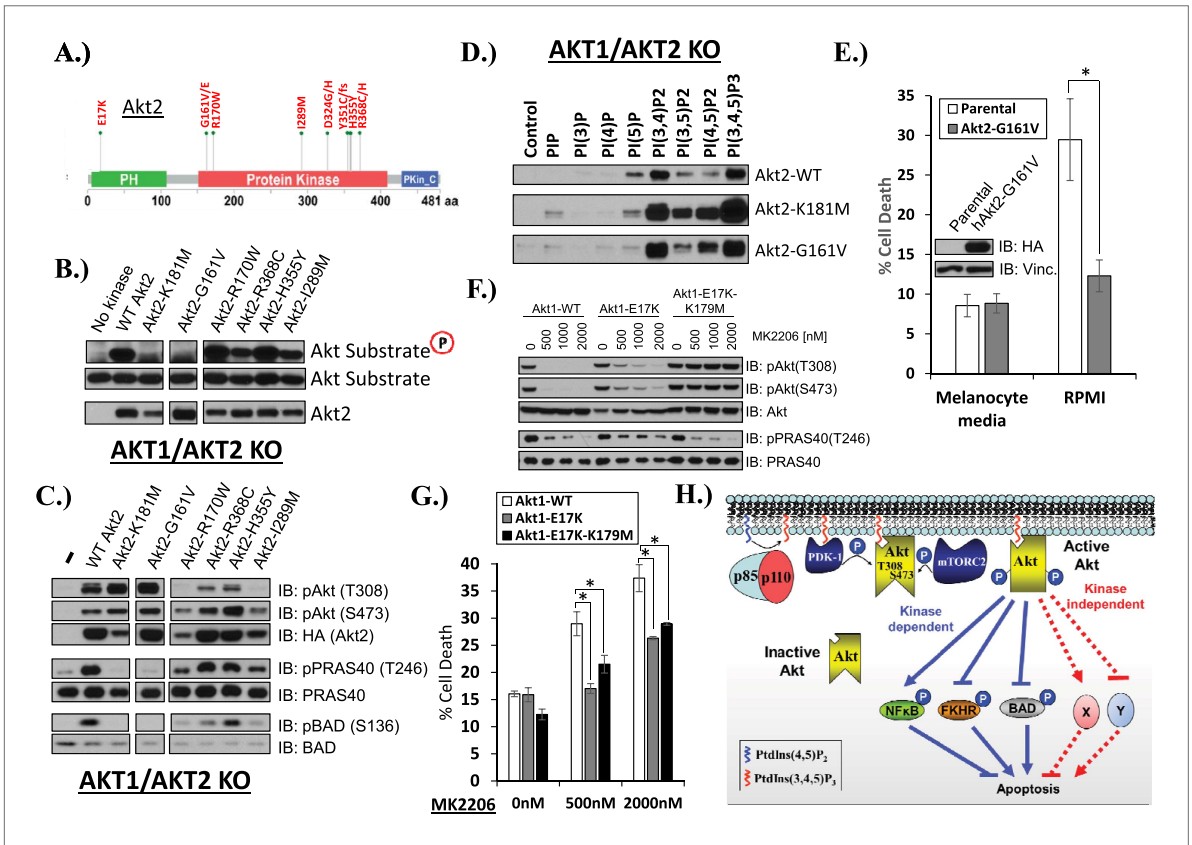

**Figure 4**. AKT mutants found in human cancer can promote cell survival independently of kinase activity. (**A**) Distribution of AKT2 mutations that occur in human cancers in 2 or more independent samples. (**B**) HA-tagged wild-type and the indicated AKT2 mutant proteins were immunoprecipitated with an HA antibody from stably transduced HCT116 AKT1−/− AKT2 −/− cells and subjected to non-radioactive in vitro kinase assay. 'No kinase' control consists of an HA-immunoprecipitate from parental HCT116 AKT1−/− AKT2 −/− cells. Substrate phosphorylation, substrate loading, and AKT2 loading were all measured by immunoblot (for further details see 'Methods'). (**C**) To evaluate the in vivo kinase activity of various AKT2 alleles, cells described in **B** were also lysed and analyzed by immunoblot with the indicated antibodies. (**D**) To determine the PIP-binding preference of WT and mutant AKT2, HCT116 AKT1−/− AKT2 −/− cells expressing WT, K181M, or G161V alleles of AKT2 were subjected to PIP-binding assay. AKT binding was assessed by immunoblot using an HA antibody. (**E**) Parental or AKT2-G161-expressing immortalized human epidermal melanocytes were plated on melanocyte media and allowed to attach overnight. Cells were then given fresh melanocyte media or switched to RMPI media containing 10% fetal bovine serum. 96 hr following media switch, cell death was assessed as elsewhere. Expression of the transgene was confirmed by immunoblot (inset). Vinculin was used as a loading control. (**F**) Parental EBC1 cells or EBC1 cells stably expressing exogenous WT AKT1, AKT1-E17K or AKT1-E17K-K179M were treated with the indicated doses of MK2206 for 24 hr and lysed. To asses target inactivation, lysates were analyzed by Western blot with the indicated antibodies. (**G**) Cells described in **F** were treated with the indicated doses of MK2206 for 96 hr. Cell death was assessed as before. (**H**) Model of AKT-dependent protection from apoptosis. AKT becomes fully activated following PI3K activation and subsequent phosphorylation at the T308 and S473 regulatory sites. Fully active AKT negatively regulates pro-apoptotic signals such as BAD and FKHR and positively regulates anti-apoptotic signals such as NFκB through phosphorylation (kinase-dependent functions). Fully AKT also regulates survival signals through kinase-independent activities.

The following source data is available for figure 4:

**Source data 1**. Contains source data for *Figure 4*.

vector and stably expressed the mutant protein in HCT116 cells with targeted deletions of AKT1 and AKT2 (HCT116-DKO). AKT2-G161V, but none of the other four kinase domain mutants, showed complete loss of in vitro AKT substrate phosphorylation, comparable to the synthetic kinase-dead *K181M-AKT2* mutant included as control in our experiments (*Figure 4B*). Whole-cell lysates from *G161V-AKT2* expressing cells showed loss of phosphorylation for the endogenous AKT kinase substrates PRAS40 and BAD (*Figure 4C*). In the phosphoinositide pull-down assay, AKT2-G161V showed altered phosphoinositide binding with acquired preference for PtdIns(4,5)P$_2$, again similar to the synthetic kinase-dead AKT2 mutant (*Figure 4D*).

Since AKT2-G161V was found in a human melanoma sample, we explored the pro-survival potential of this mutant in immortalized human melanocytes. These cells required 12-O-tetradecanoylphorbol-13-acetate (TPA) for survival in culture, as has previously been reported (*Arita et al., 1992*), but acquired the ability to survive in TPA-deficient media (RPMI) after stable expression of AKT2-G161V (*Figure 4E*, *Figure 4—source data 1*).

The increased PI(4,5)P$_2$ binding of the kinase-dead *G161V-AKT2* mutant (*Figure 4D*) was reminiscent of the most common somatic AKT mutation in human cancer (*E17K-AKT1)* which localizes to the plasma membrane due to increased affinity for the constitutive plasma membrane lipid PI(4,5)P$_2$ (*Carpten et al., 2007*; *Landgraf et al., 2008*). This raised the question whether kinase activity might be dispensable for the pro-survival functions of the oncogenic E17K-AKT1 mutant. To explore this question, we compared the effects of *E17K-AKT1* and a kinase-defective allele of this mutant (*E17K/K179M-AKT1*) in EBC1 cells. Compared to cells transduced with wild-type AKT, cells transduced with the E17K-AKT1 mutant showed residual AKT phosphorylation at equimolar concentrations of MK2206 (*Figure 4F*), consistent with the reported reduced sensitivity of *E17K-AKT1* to allosteric ATP kinase inhibitors (*Calleja et al., 2009*; *Wu et al., 2010*; *Parikh et al., 2012*). Expression of E17K-AKT1 provided partial protection from MK2206-induced cell death (*Figure 4G*, *Figure 4—source data 1*). Interestingly, the *E17K/K179M-AKT1* double mutant provided a similar degree of protection as the *E17K-AKT1* single mutant, suggesting that this cancer-associated PH-domain mutant can mediate survival signals independent of its catalytic activity in certain cellular contexts.

Lastly, we identified a human endometrial cancer cell line with a kinase domain mutation in AKT1 (G311D) (*Barretina et al., 2012*) that failed to induce phosphorylation of endogenous AKT substrates in AKT1/2 DKO HCT116 cells (*Figure 5A*). While knockdown of endogenous AKT1 or AKT2 resulted in growth inhibition, knockdown of AKT1 resulted in substantially greater cell death induction than knockdown of AKT2 (*Figure 5B*, *Figure 5—source data 1*), suggesting a state of addiction to signals provided by the mutant AKT1.

## Discussion

The work presented here represents the first example of a kinase-independent function of AKT (*Figure 4H*). Our findings that catalytically inactivating kinase domain mutations (AKT1-K179M, AKT2-K181M, AKT2-G161V) or ATP-competitive inhibitor binding to wild-type AKT can promote phosphoinositide binding document the critical importance of the kinase domain conformation on AKT PH-domain functions. It also suggests that hyper-phosphorylation of AKT in response to ATP-competitive inhibitors (*Okuzumi et al., 2009*) may not only be due to impaired phosphatase access (*Chan et al., 2011*) but also due to increased phosphorylation at the plasma membrane. Furthermore, given that the loss of phosphoinositide binding does not completely abolish the pro-survival function of kinase-deficient AKT, we suspect that the PH-domain promotes cell survival through regulation of interacting protein partners, and that perturbations in the kinase domain can influence PH-domain function by altering the ability of AKT to reach the appropriate sub-cellular compartment (likely those richest in PI(4,5)P2) and/or interact with specific effector proteins. Further studies are required to identify the effectors of catalytically inactive AKT and define its contribution to other AKT-regulated cellular processes.

Our findings have implications for the development of drugs to block AKT in cancer. Previous studies have shown that cancer-associated AKT mutations that favor an open conformation show reduced sensitivity to allosteric AKT inhibitors but retain sensitivity to ATP-competitive inhibitors (*Calleja et al., 2009*; *Wu et al., 2010*; *Parikh et al., 2012*). Here, we show that cancer cells with aberrant AKT activation by common PI3K pathway lesions (HER2, PIK3CA, MET) show the opposite drug sensitivity profile with reduced sensitivity to ATP-competitive AKT inhibitors. The inability of ATP-competitive inhibitors to suppress non-catalytic AKT functions in certain genetic contexts may represent a liability compared to allosteric AKT inhibitors. This concept remains to be tested in ongoing clinical trials.

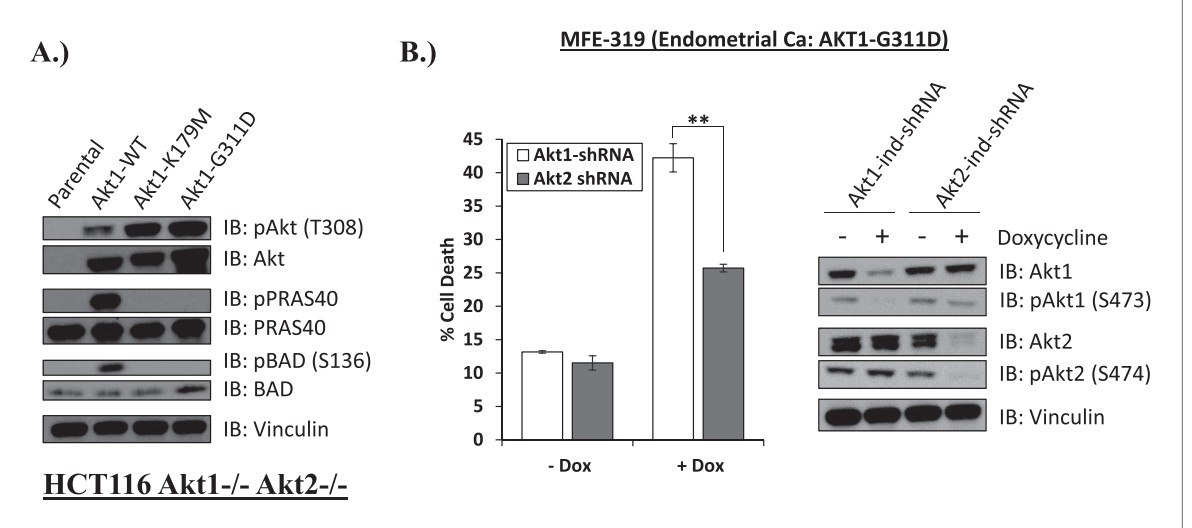

**Figure 5**. Endometrial cancer cells expressing endogenous kinase-deficient AKT1 are sensitive to AKT1 but not AKT2 knockdown. (**A**) AKT-deficient HCT116 cells were stably transduced with wild-type or the indicated mutant AKT1 alleles. In cell AKT kinase activity was determined by assessing the phosphorylation of the AKT substrates PRAS40 and BAD by immunoblot as shown. Expression of vinculin was used as a loading control. Note that AKT1-G311D does not promote AKT substrate phosphorylation. (**B**) MFE-319 endometrial cancer cells which carry and endogenous AKT1-G311D mutation (http://www.broadinstitute.org/ccle/home) were engineered to express tetracycline-inducible hairpins targeting either AKT1 or AKT2. Cells were grown in the presence or absence of 2.5 μg/ml doxycycline for 6 days. Cell death was assessed on day 7 as before (left). To confirm knockdown of AKT and selectivity of the hairpins, a separate set of plates was grown with or without doxycycline for the indicated times and lysed. Vinculin expression was used as a loading control.

The following source data is available for figure 5:

**Source data 1**. Contains source data for *Figure 5*.

## Materials

### Chemicals and antibodies

MK2206, GSK690693, and crizotinib were purchased from Selleck Chemicals (Houston, TX). GDC0068 was purchased from Active Biochem. Antibodies for immunoblots against pPRAS40 (T246), PRAS40, AKT, AKT1, AKT2, pAKT1 (S473), pAKT2 (S474), pAKT (T308), pAKT (S473), pAKT substrate (RXRXXS*/T*), BAD, pBAD, cleaved PARP, pMET (Y1349), Met, pErk1/2 (T202/Y204), Erk, pGSK3β (S9), GSK3β, pFoxo3a (S253), and Foxo3a were purchased from Cell Signaling Technologies (Danvers, MA, USA). HA, vinculin, and β-actin antibodies for immunoblots were purchased from Sigma-Aldrich (St. Louis, MO). HA.11 Clone 16B12 monoclonal antibody for immunofluorescence was purchased from Covance (Princeton, NJ). DAPI, Alexa-488 anti-mouse secondary antibody, Phalloidin 546, and Prolong Gold Antifade reagent were purchased from Life Technologies (Carlsbad, CA). HA-Tag antibody Sepharose bead conjugate and non-radioactive AKT kinase assay kit were purchased from Cell Signaling Technologies. PIP Beads Sample pack was purchased from Echelon Biosciences Incorporated (Salt Lake City, UT).

### Cells

EBC1 cells were obtained from the Japanese Collection of Research Bioresources (JCRB) Cell Bank. GTL-16 cells were the generous gift of Dr Silvia Giordano (Instituto di Candiolo, Italy). HCT116 AKT1−/− AKT2 −/− (HCT116-DKO) cells were kindly provided by Dr Bert Vogelstein (Johns Hopkins University School of Medicine). BT474, H1648, H1975, HCC827, HCC4006, MCF7, MCF10A, and MDA-MB-361 cells were purchased from the American Type Culture Collection (ATCC). Human epidermal melanocytes were purchased from ScienCell Research Laboratories (Carlsbad, CA, Catalog #2200). HCT116-DKO cells were maintained in McCoy's 5a Medium supplemented with 10% fetal bovine serum (FBS). MCF10A cells were maintained in Mammary Epithelial Cell Growth Medium (Lonza, Allendale, NJ). Human melanocytes were maintained in

melanocyte media (ScienCell, Catalog #2201). All other cell lines were maintained in DMEM media supplemented with 10% FBS.

## Plasmids and mutagenesis

TRIPZ tetracycline-inducible lentiviral shRNAs targeting human AKT1 and AKT2 were purchased from Thermo Scientific (Lafayette, CO) (AKT1 clones: V3THS_358718 and V3THS_358718; AKT2 clones: V2THS_237948, V3THS_325553, and V3THS_325558). pLNCX1-HA-mAKT1 (murine) was generated by Dr Morris Birnbaum and obtained from Addgene (Plasmid #15990). pLNCX-HA-mAKT1-K179M was generated by site-directed mutagenesis. All site-directed mutagenesis was carried out using the QuikChange Lightning Site-Directed Mutagenesis Kit from Agilent Technologies (Santa Clara, CA) and verified by Sanger sequencing. pLNCX1-ΔPH-Myr-AKT1-K179M was generated by site-directed mutagenesis using pLNCX1-ΔPH myr-AKT1 (generated by Dr Morris Birnbaum and obtained from Addgene, plasmid #15989) as template. pLNCX1-HA-AKT2 (human #27295) was generated by Dr Morris Birnbaum and obtained from Addgene (Plasmid #15990). From pLNCX1-HA-AKT2 a HindIII/ClaI restriction fragment was used to subclone AKT2 into pLPCX (Clontech) to create pLPCX-HA-AKT2. All AKT2 mutants used in this study were generated by site-directed mutagenesis using pLPCX-HA-AKT2 as template. pLNCX-HA-AKT1 (human) was generated by Dr William Sellers and obtained from Addgene (Plasmid #9004). pLNCX-HA-AKT1-K179M and pLNCX-HA-AKT1-W80A were generated by site-directed mutagenesis using pLNCX-HA-AKT1 as template. pbabe-hTERT+p53DD was generated by Dr Christopher Counter and obtained from Addgene (Plasmid #11128).

Site directed mutagenesis was done using the following primers:

AKT1-E17K (human): Sense: 5′-gctgcacaaacgagggaagtacatcaagacctg-3′, Antisense: 5′-caggtcttgatgtacttccctcgtttgtgcagc-3′.
mAKT1-K179M (murine): Sense: 5′-ccgctactatgccatgatgatcctcaagaaggagg-3′, Antisense: 5′-cctccttcttgaggatcatcatggcatagtagcgg-3′.
AKT1-K179M (human): Sense: 5′-ccgctactacgccatgatgatcctcaagaaggaag-3′, Antisense: 5′-cttccttcttgaggatcatcatggcgtagtagcgg-3′.
AKT1-R25C (human): Sense: 5′-aagacctggcggccatgctacttcctcc-3′, Antisense: 5′-ggaggaagtagcatggccgccaggtctt-3′.
AKT1-R86A (human): Sense: 5′-cagtggaccactgtcatcgaagccaccttccatgtg-3′, Antisense: 5′-cacatggaaggtggcttcgatgacagtggtccactg-3′.
AKT1-W80A (human): Sense: 5′-tccgctgcctgcaggcgaccactgtcatcg-3′, Antisense: 5′-cgatgacagtggtcgcctgcaggcagcgga-3′.
AKT2-K181M (human): Sense: 5′- gctactacgccatgatgatcctgcggaagga-3′, Antisense: 5′- tccttccgcaggatcatcatggcgtagtagc-3′.
AKT2-G161V (human): Sense: 5′-caaactccttggcaaggtaacctttggcaaagtca-3′, Antisense: 5′-tgactttgccaaaggttaccttgccaaggagtttg-3′.
AKT2-R170W (human): Sense: 5′-aaagtcatcctggtgtgggagaaggccactg-3′, Antisense: 5′-cagtggccttctcccacaccaggatgactttt-3′.
AKT2-R368C (human): Sense: 5′-tcatggaagagatctgcttcccgcgcacg-3′, Antisense: 5′-cgtgcgcgggaagcagatctcttccatga-3′.
AKT2-H355Y (human): Sense: 5′-cttctacaaccaggactatgagcgcctcttcgagc-3′, Antisense: 5′-gctcgaagaggcgctcatagtcctggttgtagaag-3′.
AKT2-I289M (human): Sense: 5′-tggacaaagatggccacatgaagatcactgactttg-3′, Antisense: 5′-caaagtcagtgatcttcatgtggccatctttgtcca-3′.

# Methods

## Western blots

Cells were harvested directly on tissue culture plates with 1% triton cell lysis buffer (Cell Signaling Technologies) supplemented with protease and phosphatase inhibitors. Lysates were sonicated, cleared by centrifugation, and normalized to equal amounts of total protein using the DC protein assay (Biorad).

## Retroviral infections

For transduction of wild-type and mutant AKT alleles into EBC1, HCT116 AKT1−/− AKT2 −/−, MCF7, and MDA-MB-361 cells, amphotropic retroviruses were generated by co-transfection of retroviral

constructs carrying the appropriate cDNA and the packaging plasmid pCL-Ampho (IMGENEX, San Diego, CA) into 293T cells. Viral particles were collected after 36 and 60 hr following transfection and target cells were infected for 18 hr with the respective virus. Stable expressors were derived through antibiotic selection. To immortalize human epidermal melanocytes (iHEM), primary melanocytes (ScienCell) were retrovirally infected with amphotropic pbabe-hTERT + p53DD virus using the above transduction protocol. For lentiviral transduction of Tet-inducible shRNAs targeting AKT1 and AKT2, viral particles were produced by co-transfection of shRNA constructs (listed above) with two packaging plasmids (pMD2G and pPAX2) into 293T cells. Five different viruses were generated (2 AKT1 hairpins and 3 AKT2 hairpins). Viral particles were collected at 36 and 60 hr after transfection. All five viruses were mixed, and infection of EBC1 cells was carried out for 8 hr after each virus collection. 36 hours after the last infection, cells were selected with 10 µM puromycin to derive stable expressors. Following selection, cells were also transduced with pLNCX-HA-mAKT1-K179M retrovirus and selected with 1 mg/ml G148 to generate cells with a hairpin-resistant AKT1-K179M allele.

## Cell death assay

Cells were seeded on 6 cm dishes and allowed to attach overnight. Cells were then treated with the indicated drugs at the indicated doses. Each treatment group was seeded in triplicate. Following treatment, both attached and unattached cells were harvested and counted on a ViCell Cell Viability analyzer. The instrument uses trypan blue to assess cell death. Cell death was expressed as the fraction of trypan blue-positive cells over the total number of cells. Statistical significance was assessed by one-tailed unequal variance t-test (ns = $p > 0.05$; * = $p \leq 0.05$; ** = $p \leq 0.01$, *** = $p \leq 0.001$).

## AKT kinase assay

AKT kinase assays were carried out using the non-radioactive AKT Kinase Assay Kit from Cell Signaling Technologies (Catalog #9840) according to the manufacturer's instructions with the following modification: HA-Tag antibody Sepharose bead conjugate was used to isolate AKT kinase. Briefly, HCT116 AKT1/2 DKO cells expressing exogenous HA-tagged AKT alleles were lysed in 1× lysis buffer and sonicated. Sonicated lysates were normalized to equal amounts of total protein. AKT was then purified from normalized lysates by HA immunoprecipitation for 3 hr at 4°C. HA beads were then washed two times with 500 µl lysis buffer and once with 1× kinase buffer. Beads were then resuspended in 50 µl kinase buffer, and kinase reaction was carried out for 30 min after addition of 1 µg of substrate and ATP to a final concentration of 200 µM. Reaction was terminated by adding 3× Laemmli buffer. 10 µl of kinase reaction were analyzed by immunoblot. Substrate phosphorylation and AKT content in each reaction was measured by immunoblot.

## PIP binding assay

HCT116 AKT1−/− AKT2 −/− (with or without exogenous AKT) cells were grown on a monolayer, washed once with cold PBS, and harvested on the dish using lysis buffer (10 mM HEPES pH 7.4, 150 mM NaCl, 1% Triton, and EDTA-free protease and phosphatase inhibitors). Lysates were sonicated and cleared by centrifugation. Cleared lysates were then normalized to equal amounts of total protein. The amount of AKT in each lysate was then quantified by Western blot using an HA antibody and the lysates normalized for AKT content using image analysis of AKT immunoreactive bands. Lysate from parental HCT116 AKT1−/− AKT2 −/− cells was used to normalize a second time for total protein amount. Each lysate was then divided into 10 aliquots to verify the final normalization by Western blot and for PIP binding analysis. PIP beads were obtained from Echelon Biosciences Inc. (Echelon Inc, Salt Lake City, UT). Each lysate was incubated with one PIP variant (30 µl of 50% bead slurry equivalent to approximately 200 pmoles) for 3 hr at 4°C. Beads were washed three times with 10 vol of lysis buffer. Bound proteins were eluted by boiling PIP beads in 3× Laemmli Sample Buffer at 95°C.

## Immunofluorescence

Cells were washed with 1× PBS at room temperature, fixed in 4% PFA for 10 min, and permeabilized for 2 min in 1× PBS containing 0.1% Triton X-100. Coverslips were blocked with 3% bovine serum albumin (wt/vol) and 0.1% Triton X-100 in PBS for 30 min. Primary antibody incubation was done for 1 hr in blocking solution, followed by five 1× PBS, 0.1% Triton X-100 washes. Secondary antibody incubation was carried out for 1 hr followed by five washes and incubated with DAPI for 2 min to visualize DNA. Fluorescent images were acquired on an upright microscope (Axio imager; Carl Zeiss)

equipped with 100× oil objectives, NA of 1.45, a camera (ORCA ER; Hamamatsu Photonics), and a computer loaded with image-processing software (Axiovision).

For confocal analysis, EBC1 cells were plated on fibronectin coated coverslips, washed with 1× PBS, fixed in 4% PFA for 10 min at room temperature, and permeabilized with 0.1% Triton X-100 for 10 min. Coverslips were then incubated in blocking solution (10% donkey serum, 0.1% BSA in PBS) for 1 hr, followed by incubation with a primary antibody (anti-HA) in blocking solution for 1 hr. Alexa-488 anti-mouse antibodies were used for detection. Coverslips were then incubated with Phalloidin 546 for 30 min. Nuclei were stained with DAPI for 3 min. Coverslips were then mounted using Prolong Gold Antifade reagent and images were acquired with a 40× magnification lens in a Leica TCS-SP5 (WLL) multiphoton confocal microscope.

## Acknowledgements

We thank members of the Mellinghoff Laboratory, Dr Charles Sawyers, Dr Neal Rosen, and Dr Michelle Garrett for helpful suggestions. This work was supported by the National Institutes of Health (1R01NS080944–01, U54CA143798, F32 GM096558), the James S McDonnell Foundation, and National Brain Tumor Society Defeat GBM initiative.

## Additional information

### Funding

| Funder | Grant reference number | Author |
| --- | --- | --- |
| National Institutes of Health | 1R01NS080944–01, U54CA143798 | Igor Vivanco, Zhi C Chen, Nicolas Yannuzzi, Carl Campos |
| National Institutes of Health | F32 GM096558 | Barbara Tanos |
| James S. McDonnell Foundation | | Igor Vivanco, Zhi C Chen, Nicolas Yannuzzi, Carl Campos |
| National Brain Tumor Society | | Igor Vivanco, Zhi C Chen, Nicolas Yannuzzi, Carl Campos |

The funders had no role in study design, data collection and interpretation, or the decision to submit the work for publication.

### Author contributions

IV, Conception and design, Acquisition of data, Analysis and interpretation of data, Drafting or revising the article; ZCC, BO, W-YH, NY, CC, Acquisition of data, Analysis and interpretation of data; BT, Conception and design, Acquisition of data, Analysis and interpretation of data; IKM, Conception and design, Analysis and interpretation of data, Drafting or revising the article

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
