## [Decision Letter]

Thank you for sending your work entitled and Senior, Thank you for sending your work entitled “A kinase-independent function of AKT promotes cancer cell survival” for consideration at eLife. Your article has been favorably evaluated by Tony Hunter (Senior Editor) and 3 reviewers, one of whom is a member of our Board of Reviewing Editors.

The Reviewing and the other reviewers discussed their comments before we reached this decision, The Reviewing editor and the other reviewers discussed their comments before we reached this decision, and the Reviewing editor has assembled the following comments to help you prepare a revised submission.

Previous studies have established that AKT provides important survival signals mediated by the phosphorylation of protein substrates, Previous studies have established that AKT provides important survival signals mediated by the phosphorylation of protein substrates. Here the authors demonstrate that AKT also promotes survival by a mechanism that is independent of AKT substrate phosphorylation. The identity of this mechanism is not described. Nevertheless, the conclusions of the paper are novel, interesting, and potentially significant for cancer therapy. However, the conclusions of this study are not fully supported by the data presented.

Major points:

1) One weakness of this study is that it is largely based on over-expression of mutant AKT. Controls using WT AKT are not presented (with the exception of Figure 4); this important control should be included in each experiment. Moreover, studies of AKT expressed at endogenous levels are essential. Only two studies designed to examine AKT expressed at endogenous levels are presented. The first is a shRNA knock-down/rescue with a shRNA-resistant mutant (Figure 2). This experiment is difficult to interpret because the mutant AKT is clearly over-expressed and knocked down by the shRNA (despite being stated to be shRNA-resistant). The experiment also lacks controls for off-target actions of the shRNA pools. The second study is an shRNA experiment on endometrial cancer cells (Figure 5) that lacks controls for off-target effects of the shRNA pools. Additional validation of a kinase-independent function of AKT expressed at endogenous levels is required. For example, can the survival function of kinase-dead AKT be phenocopied by blocking upstream pathways that lead to AKT activation?

2) Figure 2: The authors argue that the ability of ATP-competitive AKT inhibitors to create complexes with their target(s) that are capable of providing survival signals is revealed by their ability to suppress crizotinib-induced cell death. This would be more clearly validated by a comparison with an “allosteric” inhibitor.

3) The protein localization data are unclear (Figure 3), because the conclusions regarding membrane localization are compromised by the apparent membrane ruffling that causes a higher local density of membrane that can give the false appearance of concentration.

4) Figure 3: The data obtained with mAKT1-K179M lacking a PH domain are difficult to interpret. Is it targeted to the plasma membrane? Is it substantially resident in other membrane compartments? Could these data indicate that membrane targeting is important, rather than indicating that the survival signals are generated away from membranes, as stated?

5) The analysis is focused on cells with gain-of-function of pathways leading to Akt activation, including MET and PI3K-pathway mutations. Data on survival in response to allosteric vs ATP-competitive AKT inhibitors using cells without this gain-of-function would provide some initial insight into the potential cancer v normal-cell therapeutic window.

6) The HA blot in Figure 3—figure supplement 2 is difficult to interpret.

---

## [Author Response]

In response to the reviewers’ excellent suggestions, we have conducted a series of additional experiments that clarified several key points brought up by several reviewers:

To address the role of AKT protein levels in our observations with kinase-dead AKT allels, To address the role of AKT protein levels in our observations with kinase-dead AKT allels, we now expressed wildtype AKT in multiple cell lines (EBC1, MCF7, MDA-MB-361) and show that wild-type AKT is not sufficient to promote cell survival. We also provided more experimental detail regarding our experiment where we knocked down endogenous (human) AKT and replaced it with a murine, kinase dead AKT protein.

Regarding the subcellular localization of kinase-deficient AKT mutants, Regarding the subcellular localization of kinase-deficient AKT mutants, we performed additional confocal microscopy experiments. We again found increased plasma membrane localization of kinase-deficient AKT. Furthermore, by re-introducing wild type or kinase-deficient AKT alleles into AKT1/2 double knockout cells, we show enhanced activation loop phosphorylation (T308) in mutant AKT, a phosphorylation event that occurs at the cytoplasmic membrane.

We now document that the allosteric AKT inhibitor, We now document that the allosteric AKT inhibitor, unlike ATP-competitive AKT inhibitors, does not antagonize the ability of a MET kinase inhibitor to induce cell death in MET-amplified cancer cells. This important control was missing from our original manuscript.

Major points:

1) One weakness of this study is that it is largely based on over-expression of mutant AKT. Controls using WT AKT are not presented (with the exception of Figure 4); this important control should be included in each experiment.

We thank the reviewers for this suggestion and agree that assessing the effects of ectopically expressed wild type AKT represents an important control, We thank the reviewers for this suggestion and agree that assessing the effects of ectopically expressed wild type AKT represents an important control. To address this question, we now overexpressed wild type AKT1 or wild type AKT2 in several cancer cell lines and repeated key experiments with this additional control. Unlike the W80A-AKT mutants, which cannot bind MK2206, expression of wild type AKT1 (Figure 6) or AKT2 (Figure 6) does not rescue MDA-MB361 breast cancer cells from MK2206-induced cell death (indicated by red arrows below). We made similar observation in EBC1 lung cancer cells (Figure 6). These new data are included in our revised manuscript as Figure 1, Figure 1—figure supplement 5, Figure 2—figure supplement 1, and Figure 4 (also shown below as Figure 6, respectively).Author response image 1.A) and B) MDA‐MB‐361 cells were stably transduced with wild type or MK‐2206‐resistant AKT1 (AKT1‐W80A) or AKT2 (AKT2‐W80A) and assessed for MK2206 response as shown. Note that overexpression of wild type AKT alleles does not change the response to MK2206. C) EBC1 cells were stably transduced with wild type or kinase‐dead AKT2 (AKT2‐K181M) and assessed for MK2206 response as in A. Note that mutant, but not wild type AKT overexpression, confers resistance to MK2206.

Moreover, studies of AKT expressed at endogenous levels are essential. Only two studies designed to examine AKT expressed at endogenous levels are presented. The first is a shRNA knock-down/rescue with a shRNA-resistant mutant (Figure 2). This experiment is difficult to interpret because the mutant AKT is clearly over-expressed and knocked down by the shRNA (despite being stated to be shRNA-resistant). The experiment also lacks controls for off-target actions of the shRNA pools. The second study is an shRNA experiment on endometrial cancer cells (Figure 5) that lacks controls for off-target effects of the shRNA pools. Additional validation of a kinase-independent function of AKT expressed at endogenous levels is required. For example, can the survival function of kinase-dead AKT be phenocopied by blocking upstream pathways that lead to AKT activation?

We thank the reviewers for the opportunity to clarify these important points.

The reviewer is correct that the shRNA-“resistant” cDNA is, in fact, only partially resistant to our shRNAs. The targeting sequences used to knock down AKT1 in our experiments (CCCGAGGTGCTGGAGGACA and GGCAAGGTGATCCTGGTGA) are based on human AKT1 and we used murine AKT as a “rescue” allele. As indicated in the Materials and methods section of our revised manuscript, human and murine AKT1 are mismatched only at one or two nucleotides (underlined) relative to each of the targeting sequences (CCTGAGGTGCTGGAGGACA and GGGAAGGTGATTCTGGTGA), likely explaining the incomplete rescue. Nonetheless, despite the effects of the shRNA-hairpins on the murine mRNA, only cells expressing the murine AKT1 cDNA show residual AKT1 following doxycycline treatment. [Please compare, in Figure 7 below, lanes 6 and 8 (green arrows) to lanes 2 and 4 (red arrows).] Of note, levels of residual K179M-mAKT1 (lanes 6 and 8, green arrows) are roughly equivalent to the endogenous AKT levels seen in the absence of doxycycline (lanes 1 and 3, blue arrows), an effect not seen with AKT2 expression. In our revised manuscript, we have revised the figure and text to explain this point more clearly.Author response image 2.EBC1 cells were stably transduced with doxycycline‐inducible human‐specific AKT1 and AKT2 shRNAs (Control) and subsequently transduced with a murine kinase‐deficient AKT1 cDNA (mAKT1‐K179M). Cells were treated with doxycycline for the indicated times. AKT levels were measured at each time point by western blot as shown. Note that the levels of shRNA‐resistant AKT1 expression (green arrows) are equivalent to those seen in untreated control cells (blue arrows).

We also examined the question whether the survival function of kinase-dead AKT can be phenocopied by blocking upstream pathways that lead to AKT activation. We found that the ATP-competitive AKT inhibitor GSK690693 does not promote cell survival if we block all AKT activating signals through pharmacological PI3-kinase inhibition with the PI3K inhibitor BKM-120 (Figure 8). These data suggests that even the kinase-independent functions of AKT require a PI3K input. We are therefore unable to experimentally phenocopy the pro-survival functions of kinase-deficient AKT by blockade of upstream signals.Author response image 3.EBC1 cells were treated with the pan‐class‐I PI3K inhibitor, and/or the AKT inhibitor GSK690693. Cells were allowed to grow for 96 hours and assessed for cell death following this incubation period by the trypan blue method (left). A duplicate set of samples was analysed by western blot to document drug activity. Note that GSK690693 does not protect (red arrow) from PI3K‐inhibitor‐induced cell death (green arrow).

2) Figure 2: The authors argue that the ability of ATP-competitive AKT inhibitors to create complexes with their target(s) that are capable of providing survival signals is revealed by their ability to suppress crizotinib-induced cell death. This would be more clearly validated by a comparison with an “allosteric” inhibitor.

We apologize for not including these data in our prior submission. In our revised manuscript, we are showing the full experiment which included the allosteric AKT inhibitor MK2206 and the crizotinib/MK2206 combination. The results show that the allosteric AKT inhibitor MK2206, unlike the ATP-competitive inhibitor GSK690693, does not protect from crizotinib-induced cell death (Figure 9), consistent with our model. These data are included as Figure 2 in the revised manuscript.Author response image 4.EBC1 cells were treated with the MET inhibitor crizotinib and/or AKT inhibitors GSK690693 and MK2206, and assessed for cell death. Panel (A) was shown in the original manuscript as Figure 2 and only contained data with crizotinib and GSK690693. Panel (B) has now replaced this figure in the revised manuscript and contains additional data using MK2206 (red arrows).

3) The protein localization data are unclear (Figure 3), because the conclusions regarding membrane localization are compromised by the apparent membrane ruffling that causes a higher local density of membrane that can give the false appearance of concentration.

We thank the reviewers for the opportunity to clarify this important point. In our original manuscript, we showed increased phosphorylation of the kinase-deficient AKT1 mutant (K179M) at Serine 308 (Figure 10). Other investigators have shown that phosphorylation of AKT at T308 occurs at the plasma membrane (Stephens et al., Science 1998, PMID: 9445477), and we believe that this is consistent with our previously provided immunofluorescence data (Figure 10).

In response to the reviewers, we have now also examined the cellular localization of kinase deficient AKT2 (K181M). As shown in confocal images of wild type vs. mutant AKT2 in stably-transduced EBC1 cells (Figure 10, left column), kinase-deficient AKT2 more robustly colocalizes with the plasma membrane marker Na+/K+ ATPase than its wild type counterpart. Furthermore, following the reviewer’s suggestion, we co-stained these cells with phalloidin which binds F-actin and marks membrane ruffles. We found increased membrane localization of mutant AKT2 even in the absence of membrane ruffles (red arrows in Figure 10, right column). These data are included in Figure 3—figure supplement 2 of our revised manuscript.Author response image 5.MCF10A cells were stably transduced with wild type or kinase‐dead (K179M) AKT1. Western blot analysis (A) (Figure 3 of revised manuscript) shows enhanced pAKT(T308) in mutant‐AKT‐expressing cells, and immunofluorescence analysis indicates this increase corresponds to enhanced membrane localization (B) (Figure 3 of revised manuscript). (C) EBC1 cells were stably transduced with WT or kinase‐dead AKT2 (K181M). Cells were stained as indicated and imaged using confocal microscopy. Note the enhanced membrane localization of mutant AKT2 (bottom left, green arrow), compared to wild type AKT2 (top left, green arrow). Insets are an enlarged images of the region indicated by the red arrows.

We thus conclude that kinase-deficient AKT1 and AKT2 show increased localization to the plasma membraneOur results with engineered and naturally occurring kinase-dead AKT mutants are reminiscent of the findings from Kevan Shokat’s group at UCSFwhich documented relocalization of AKT to the plasma membrane in response to ATP-competitive Akt kinase inhibitors ([17] We thus conclude that kinase-deficient AKT1 and AKT2 show increased localization to the plasma membrane. Our results with engineered and naturally occurring kinase-dead AKT mutants are reminiscent of the findings from Kevan Shokat’s group at UCSF, which documented relocalization of AKT to the plasma membrane in response to ATP-competitive Akt kinase inhibitors (Okuzumi et al., Nature Chemical Biology, 2009, PMID: 19465931). In contrast, the allosteric AKT inhibitor MK2206 does not promote membrane relocalization of AKT.

4) Figure 3: The data obtained with my-Akt1-K179M lacking a PH domain are difficult to interpret. Is it targeted to the plasma membrane? Is it substantially resident in other membrane compartments? Could these data indicate that membrane targeting is important, rather than indicating that the survival signals are generated away from membranes, as stated?

We completely agree with the reviewers’ interpretation of the data and have rephrased our conclusions to express this more clearly. The non-catalytic function of AKT requires the presence of the PH domain, as demonstrated by our data with the Δ PH-Myr-Akt1-K179M mutant. Together with the results of our experiments using the R86C and R25C mutants, this data suggest that the kinase-independent function of AKT might require targeting to specific membranes or membrane subdomains, likely those richest in PI(4,5)P2, to which the mutants show enhanced binding.

5) The analysis is focused on cells with gain-of-function of pathways leading to Akt activation, including MET and PI3K-pathway mutations. Data on survival in response to allosteric vs ATP-competitive AKT inhibitors using cells without this gain-of-function would provide some initial insight into the potential cancer v normal-cell therapeutic window.

We thank the reviewers for this insightful suggestion, We thank the reviewers for this insightful suggestion.

To address the issue, To address the issue, we have treated BEAS-2B immortalized (non-transformed) bronchial epithelial cells with MK2206, and found that it could not induce any appreciable amount of cell death (Figure 11).

We have also treated three NSCLC cell lines carrying endogenous activating EGFR mutations with AKT inhibitors (also shown in Figure 11), We have also treated three NSCLC cell lines carrying endogenous activating EGFR mutations with AKT inhibitors (also shown in Figure 11). While we have found that amplification of MET and activating PIK3CA-mutations are associated with sensitivity to AKT allosteric inhibitors, we find that the EGFR-mutant lung cancer cell lines are completely refractory to cell death induction by either MK2206 or GSK690693. Given that these EGFR mutations are capable of activating AKT (PMID: 15731348), these data suggest that the mechanism of PI3K activation (e.g. EGFR activation vs MET activation) is likely critical in establishing AKT dependence. These data have been added to the revised manuscript as Figure 1—figure supplement 4.Author response image 6.Immortalized bronchial epithelial cells (BEAS‐2B) and EGFR‐mutant non‐small‐cell lung carcinoma cell lines (HCC827, HCC4006, and H1975) do not die in response to MK2206.

6) The HA blot in Figure 3—figure supplement 2 is difficult to interpret.

We apologize for the poor quality of this blot. We have now performed additional western blot analysis with an AKT1-specific antibody to quantify AKT1 expression in the transduced sublines, relative to the parental controls. The results show increased AKT1 expression in cells transduced with wild type or mutant AKT1 cells (see image quantification below each lane in Figure 12). Of note, it also again shows that the kinase-dead K179M-AKT1 mutant exhibits increased phosphorylation at threonine 308 (lanes 5) compared to wild type AKt1 (lane 3) and that this hyperphosphorylation is lost when PIP3-binding is disrupted by the second R86A mutation (lane 7). These new data have been included as Figure 3—figure supplement 4 of the revised manuscript.Author response image 7.MDA‐MB‐361 cells were stably transduced with the indicated alleles of AKT1. Cells were treated with vehicle or MK2206 and lysed for western blot analysis as indicated. AKT1 levels and the levels of the loading control (vinculin) were quantified by image densitometry, and the relative levels are indicated under each band. Note that all samples from transduced cells express higher AKT1 levels, and that Akt1‐K179M exhibits enhanced pAKT(T308) levels.